# Incomplete Ileocecal Bypass for Ileal Pathology in Horses: 21 Cases (2012–2019)

**DOI:** 10.3390/ani11020403

**Published:** 2021-02-05

**Authors:** Gessica Giusto, Anna Cerullo, Federico Labate, Marco Gandini

**Affiliations:** Department of Veterinary Sciences, University of Turin, Largo Paolo Braccini 2-5, 10095 Grugliasco (TO), Italy; gessica.giusto@unito.it (G.G.); anna.cerullo@unito.it (A.C.); federico.labate@unito.it (F.L.)

**Keywords:** horse, colic, ileal pathology, anastomosis, ileocecal bypass

## Abstract

**Simple Summary:**

The ileal pathologies represent a problem often found during colic. At exploratory laparotomy, in cases where there is involvement of the ileum and there is a suspicion of an ileocecal valve disfunction, the surgeon may be faced with the choice of whether to resect the intestinal tract involved, do nothing, or perform an ileocecal bypass without resection of the ileum. This latter technique may represent a valid alternative to extensive manipulation, and it may reduce the recurrence of ileal occlusion and post-operative complication. This study aims to describe clinical findings, surgical techniques, and post-operative progress of horses who have undergone an incomplete ileocecal bypass in case of strangulating or non-strangulating ileum pathologies. Incomplete ileocecal bypass may represent an effective and safe surgical technique in these cases and in all cases of ileal pathologies treated without resection to avoid recurrence and reduce complications.

**Abstract:**

Background: Incomplete ileocecal bypass can be performed in cases in which an ileal disfunction is suspected but resection of the diseased ileum is not necessary. Objectives: To describe the clinical findings, the surgical technique, and the outcome of 21 cases of colic with ileal pathologies that underwent an incomplete ileocecal bypass. Methods: Historical, clinical, and surgical features of cases diagnosed with pathologies involving the ileum or the ileocecal valve that underwent ileocecal anastomosis without ileal resection were retrieved. Clinical (heart rate, duration of symptoms, presence of reflux, age, weight at arrival) and surgical (surgical pathology, duration of surgery, type of anastomosis) data were retrieved and analysed. Data on short term survival and postoperative complications (colic, post-operative reflux, incisional infection, fever), length of hospital stay, and long term follow up were also obtained. Results: A total of 21 horses met the criteria; 13 horses had ileal impaction (one with muscular hypertrophy), 5 horses had epiploic foramen entrapment, and 3 horses had a pedunculated lipoma. An incomplete ileocecal bypass was performed with a two-layer hand-sewn side-to-side technique in 19 cases and with a stapled side-to-side technique in 2 cases. Short term survival was 95.2%. At 12-months follow up, all horses but two were alive, and 13 of the 14 sport horses returned to their previous level of activity. Long term survival was 90.47%. Conclusions Incomplete ileocecal bypass may represent a valid surgical technique in case of ileocecal valve disfunction when ileum resection is not necessary; this technique may represent an alternative to extensive manipulation without subsequent recurrence of ileal impaction.

## 1. Introduction

Although colic is a frequently occurring problem in horses, most cases of colic can be resolved medically, and only a small percentage require emergency surgery to avoid fatal outcomes. Approximately 6.6% of small intestine surgical pathologies in horses involve the ileum [1]. The ileum may be involved in a simple obstruction of the ileocecal valve or, as often happens, in strangulating lesions. The reason for this is not clear, and only hypothetical explanations, such as that peristalsis draws it into the entrapment, have been provided [2,3]. In strangulating lesions, if the ileum is necrotic, there is no other choice than resection and anastomosis (either jejunocaecal or jejunoileal); in all other cases (simple obstruction or damaged but viable), it may be necessary to decide to resect the ileum and perform an incomplete ileocecal bypass or manage these conditions conservatively.

Ileocecal bypass has been proposed both in strangulating and non-strangulating diseases of the ileum by several authors [4,5,6,7] and rejected by others [8,9]. It is mainly proposed in cases of muscular hypertrophy or ileocecal intussusception [5,7], while it is not recommended by some authors for ileal impaction [8,9]. Ileal impaction has distinct causes such as Bermuda Grass Hay ingestion [10] and *Anoplocephala perfoliata* infestation [11], although in many cases, a definitive causative agent cannot be found, especially in Italy, where Bermuda Grass Hay and serological tests for *Anoplocephala* are not available, while faecal tests yield an extremely low sensitivity [12]. Attempts to define the causes of ileal impaction are therefore based on the deworming history, which is not always complete. However, it is not possible to define if the apparently viable but thickened and oedematous ileum will resume its functionality in the early postoperative period. If this condition is treated conservatively, without performing a bypass, there is a risk of the recurrence of the same problem (i.e., impaction) or the development of post-operative ileus in the early post-operative period. Therefore, in cases where risk of recurrence is suspected in the presence of a questionably functional ileum, an incomplete ileocecal bypass may be a valid alternative. A complete ileocaecal bypass can be performed in these cases, but it is associated with a high complication rate and a shorter survival [13], while in many cases, conservative treatment can be successful, especially in simple obstructions such as ileal impaction.

While several authors have mentioned incomplete ileocecal bypass, there is no report specifically analysing the outcome of such a technique. The aim of the present study was therefore to report indications, technique, complications, and outcomes of a case series of horses submitted to incomplete ileocecal bypass.

## 2. Materials and Methods

### 2.1. Case Selection

We used records of horses that underwent colic surgery at the Veterinary Teaching Hospital of Turin between January 2016 and December 2019. Records were searched by the authors for horses that were submitted to incomplete ileocecal bypass because of pathologies involving the ileum but without visible indication of non-viable intestine. Clinical (heart rate, duration of symptoms, presence of reflux, age, weight at arrival), haematological (packed cell volume (PCV), total protein (TP), venous lactate), and surgical (duration of surgery, type of anastomosis) data were retrieved and analysed. The parameter “duration of symptoms” was defined as the interval between admission to the hospital and the “last time seen normal” by the owner/stable keeper. Surgical pathology was determined by the definitive diagnosis made at surgery, and the same was considered for the type of anastomosis. Data on short term survival and possible post-operative complications (colic, post-operative reflux, incisional infection, fever), length of hospital stay, and long-term follow-up were also obtained. Short-term complications included a diagnosis of incisional infections (defined as any discharge from the wound other than for serosanguineous discharge), post-operative colic, or post-operative reflux (POR) that occurred before discharge of the horse from the hospital. Here, POR was defined as any net retrieval of fluids at nasogastric intubation which was performed only if clinical signs (increased HR, pain, increase in PCV/TP) were present. Short-term survival was defined as the horse being discharged alive from the hospital. Length of hospital stay was assessed as the post-operative day in which the horse was considered dischargeable (to avoid bias given by decision of the owners for collecting the horse).

### 2.2. Surgical Procedure

All horses received pre-operative antibiotics immediately before induction of anaesthesia (gentamicin 6.6 mg/kg IV, benzylpenicillin 12.000 UI/Kg IM), pre- and post-operative administration of flunixin meglumine (1.1 mg/kg IV bid), IV fluids (Lactate’s Ringer), ranitidine (1.5 mg/kg IV q8h), and calcium gluconate 23% (0.5 mL/kg IV bid) as needed. Anaesthesia was induced with ketamine and diazepam administration and maintained with isoflurane in oxygen administration. All horses received dexamethasone 0.05 mg/kg IV before making the ileal bypass. After resolution of the obstruction, a thorough examination of the small intestine and the stomach with an incomplete ileocecal bypass was performed when there was a suspicion of dysfunction but not vascular compromise of the ileum and the ileocecal valve, combined or not with the presence of large quantities of solid material in the stomach. A small intestinal enterotomy was performed in those cases where resolution of the ileal impaction required extensive manipulation. The site for enterotomy was either on the jejunum or on the ileum. Whenever possible, the enterotomy site was used as ileal stoma for the bypass procedure. To obtain this, we checked that the proposed enterotomy site on the ileum could be brought far enough from the abdominal incision on the right side of the horse to allow safe emptying of its content. After further draping, a full thickness incision was performed on the antimesenteric side of the ileum, and its content was emptied out of the surgical field into a bucket. Then, a two-layer, handsewn, side-to-side anastomosis was performed.

#### 2.2.1. Hand-Sewn Technique

The operating field was isolated with absorbent surgical drapes to avoid contamination. The antimesenteric side of the jejunum was attached with two stay sutures to the body of the cecum midway between the dorsal and the medial band (Figure 1).

Subsequently, a two layer handsewn side-to-side anastomosis without resection was performed to complete the bypass. The assistant surgeon placed tension on the stay sutures on the same side to align the two intestinal tracts. The first layer of the far side of the anastomosis obtained was with a continuous Lembert suture of USP 2-0 polydioxanone (PDS), joining the antimesenteric side of the ileum and the caecal wall. Once the suture was completed, a 10–12 cm serosubmucosal incision was performed approximately 5 mm on both sides of the suture.

A full thickness continuous suture of USP 2-0 PDS comprising mucosa and submucosa of each intestinal segment was applied to complete the underside of the anastomosis (Figure 2).

The mucosa of the two intestinal segments was then incised to create the stoma. Subsequently, a full thickness simple continuous suture joining the edges of the incisions on the ileum and the cecum completed the inner layer of the near side of the anastomosis. A second layer of a Cushing suture of USP 2-0 PDS was applied to complete the anastomosis (Figure 3).

#### 2.2.2. Stapled Technique

As for the previous technique, the antimesenteric side of the jejunum was attached to the body of the cecum midway between its dorsal and its medial band with two stay sutures with USP 2-0 PDS. A small enterotomy was performed in both the jejunum and the cecum wall to allow insertion of the two arms of the linear cutting stapler (GIA80, Medtronic Italia SpA, Milano, Italy). The stapler was closed and fired, and the two lines of staples were deployed on the ventral and the dorsal edge of the stoma, which was created simultaneously by a stapler blade between the two staple rows. The enterotomies were closed with a continuous Cushing pattern of USP 2-0 PDS, and the staple line was not oversewn. In both techniques, the anastomosis was tested, and the two intestinal segments were repositioned into the abdomen. At the end of the surgery, the abdomen was closed routinely with a double-layer closure; the *linea alba* was closed with a simple continuous pattern of USP 2 PDS, and the skin was closed with a continuous horizontal mattress suture pattern with USP 1-Nylon. All the horses recovered from anaesthesia uneventfully.

In both techniques, the space among the bypass and the ileocecal valve was occluded by suturing the ileocecal fold on itself in order to obliterate a possible internal hernia foramen.

## 3. Results

### 3.1. Case Signalment

Twenty-one cases were included in this study, referred to the Veterinary Teaching Hospital of Turin between 2012 and 2019. Of the 21 horses, 13 were geldings, 5 were mares, and 3 were stallions. Their age ranged from 2 to 19 years (mean 10 years), and their breeds included Warmblood (*n* = 5), Italian Saddlebred (*n* = 4), Selle Francais (*n* = 3), American Quarter Horses/Paint (*n* = 2), Standardbred (*n* = 2), ponies (*n* = 2), Andalusian (*n* = 2), and Arabian horse (*n* = 1). Their weight before surgery was 471 ± 92 kg.

### 3.2. Clinical Findings

The average duration of abdominal pain before admission was 12 h (range 6–48 h). The heart rate at admission was 56 bpm (range 40–100), PCV was 44 ± 7%, TP 7 ± 1 g/dL, and the concentration of blood lactate was 4.5 mmol/L mean (range 1–10). The mean amount of reflux was 0.4 L (range 0–4).

### 3.3. Surgical Findings

A final diagnosis of ileal impaction was made in 13 horses (62%, 8 geldings, 4 females, 1 male), epiploic foramen entrapment was found in 5 cases (24%, 2 geldings, 1 male, 1 female), and pedunculated lipoma in 3 cases (14%, 2 geldings, 1 female) during exploratory laparotomy.

The length of the intestine involved was 0.7–2 m for ileal impaction and 1–2.5 m for strangulating lesions. In all cases, the ileum was viable but thickened and oedematous. The surgery had an average duration of 125 ± 18 min. To achieve a successful resolution, 0.66 m of small intestine (range 0.5–1 m) was bypassed.

### 3.4. Post-Operative Complications

Three horses presented post-operative reflux (2 epiploic foramen entrapment, 1 lipoma), which was resolved with medical treatment in two cases. In one case, a new laparotomy was performed after 3 days, and a small intestinal volvulus was found. The volvulus had caused necrosis of a large part of the jejunum and, in agreement with the owner, the horse was euthanised. Two horses presented a moderate infection of the surgical wound with sero-purulent discharge, which was resolved with local treatment.

After a mean of 9.2 days (range 7–15), 20 out of the 21 horses were discharged alive from the hospital. A telephone follow-up was performed with the owners for long-term assessment.

About 8 months after discharge, one horse came back to the clinic with signs of abdominal pain; right dorsal displacement unresponsive to medical treatment was diagnosed. Surgery was recommended but refused by the owner, and euthanasia was performed. At necropsy, the ileo-caecal bypass appeared anatomically viable.

All remaining horses (19/21) were alive at 12 months. During this period, one horse had developed three episodes of colic that were resolved medically; 13 out of 14 sport horses returned to their previous levels of sport activity.

## 4. Discussion

This article describes the use of ileocecal bypass to reduce post-operative complications after resolution of ileal pathologies managed without resection of the ileum. Ileal impaction is the most common non-strangulating lesion in horses and is associated with the consumption of coastal Bermuda hay [13] that is widespread in the south of the United States. In places where coastal Bermuda hay is not present, ileal impaction may be associated with the presence of tapeworm [11], which causes mucosal ulceration and oedema, resulting in a reduced lumen of the ileocecal valve and, consequently, in ileal impaction. Another cause of ileal impaction is muscular hypertrophy of the ileum, which is considered idiopathic [14]. This condition causes recurrent colic syndrome in horses and is more frequent in mature horses [15]. If surgery is required, the impaction can be broken manually by mixing it with fluid from the proximal bowel. If the impaction is severe, enterotomy is necessary. Side-to-side jejunocaecal or ileocecal anastomosis is recommended if there is a suspicion of impaired ileocecal valve function. Incomplete ileocecal bypass has a better survival rate than complete side-to-side jejunocaecal anastomosis [9], although some authors do not recommend it in simple obstructions [8].

We performed incomplete ileocecal bypass in cases of ileal impaction to avoid recurrences when the cause was not clear (the horses were fed good quality hay, and none of them had been recently tested or treated specifically in the previous 6 months for tapeworm). We hypothesised that, in many cases (7 out of 10), the cause of ileal impaction was undetected *Anaplocephala perfoliata* infestation, although specific serologic tests to detect this parasite are not available in our country. Faecal tests for *Anoplocephala perfoliata* were positive in 50% of the cases of ileal impaction reported in our study. In the other cases, we hypothesised some sort of disfunction of the ileum, not being able to relate the impaction to a particular type of hay or food. Resolution of ileal impaction with softening of the content and external manipulation is the recommended method [8]. Nevertheless, these cases tend to have recurring impaction in the short post-operative period with a high rate of POR [10]. This is probably due to the permanence of an inflamed and thickened ileocecal valve and ileal wall, needing several days to subside.

A recurrence impaction of the valve is a not a rare occurrence in these cases, especially when a stomach impaction can be palpated during surgery. In our case series, 8 out of 10 horses with ileal impaction and 4 out of the remaining 11 horses were found with a partial stomach impaction at surgery. Mobilisation of this material in the early post-operative period, concurrently with a not fully functional ileocecal valve, can be responsible for the occurrence of POR in these cases. For the same reason, we also performed an incomplete ileocaecal bypass in five cases with strangulating lesions in which the distal jejunum was involved. Thus, the ileum was considered viable but with a very thick and oedematous wall and, in some cases, scarcely motile, which we considered to be a potential predisposing factor for the development of post-operative reflux.

Since it was not possible to remove, in the immediate post-operative period, the underlying cause (ileocaecal valve thickening, parasite presence, viable but thickened intestinal wall), we opted to surgically deviate the flux of ingesta to avoid an early recurrence of the problem.

For ileal impactions in particular, leaving the intestine without performing an anastomosis could have been a viable option, while proposing a second laparotomy in case complications would arise. This is the best approach according to some authors that reported good results with this conservative approach [8,9]. The high success rate of this approach can be explained by the fact that, in the reported cases, ileal impaction was mostly caused by a specific cause, such as the consumption of Bermuda grass hay. Since this species does not occur in Italy, causes of ileal impaction in this country may have different, less specific causes. This leads to a high risk of recurrence, not being possible to fully recognise, and thus remove, the cause. Further, because of financial constraints, owners in our country rarely allow a second surgery to be performed, and surgeons must be aware that they must do everything they possibly can to reduce post-operative complications; this may eventually imply a more “aggressive” surgery. In many cases, we opted for an ileal enterotomy to empty the intestine from fluids and to reduce the impaction. Although its use is controversial [16], we believe that extensive manipulation of the intestine to resolve the impaction can be as (or more) detrimental as an enterotomy.

To perform the bypass, the stoma must be opened as close as possible to the ileocecal valve to prevent an excessive length of intestine from producing a kinking of the bypassed tract.

In many cases, we were able to place the ileal entorotomy at the site identified for the ileocecal anastomosis, thus avoiding a double incision. This method considerably shortens the surgical time and reduces the manipulation of the intestine (Figure 4). Certainly, performing the anastomosis with the lumen of the small intestine fully opened requires careful isolation of the surgical field to avoid contamination.

In the other cases, either an enterotomy was not performed or, in cases with an extensive impaction, a jejunal enterotomy was performed proximally to the site where the ileocecal anastomosis was then performed.

In all cases, a single dose of dexamethasone was also administered at surgery to resolve the inflammation and the oedema encountered in both strangulating and non-strangulating obstructions. Under this perspective, this treatment can be viewed as a possibility to reduce post-operative ileus, as recently reported [17].

In our cases, incomplete bypass was performed mostly with a two-layer hand-sewn technique, except in two cases where there was the need to reduce the surgical time after difficult resolutions of the lesion. This is quite an exceptional occurrence, since cases with ileal lesions are not the best candidates for the use of a stapling device due to the thickness of the intestinal wall. One of these cases developed post-operative complications and was submitted to a second laparotomy, where a small intestinal volvulus was found. Nevertheless, the anastomosis was viable, and no signs of leakage associated with the use of staplers were found.

In our cases, ileal impaction occurred predominantly in intact and castrated males. Although we could not find a reason for this, it is opposite to what has been reported in ithe literature, where females are the ones most found with this pathology [13].

The limitations of this study are mainly represented by the retrospective nature of it and the small number of cases, although it remains comparable to other studies on intestinal anastomosis in horses.

## 5. Conclusions

In different geographical areas, ileal impaction may have different causes and features and thus may require different treatments. While, in some cases, manipulation may be effective in resolving the disorder and in avoiding recurrence, in other cases, due to different deworming routines, feeding practices, financial issues, and surgeon experiences, a more articulated surgical approach could be needed to resolve the pathology and reduce complications.

Incomplete ileocecal bypass may represent an effective and safe surgical technique in these cases and in all cases of ileal pathologies treated without resection to avoid recurrence and reduce complications.

## Figures and Tables

**Figure 1 animals-11-00403-f001:**
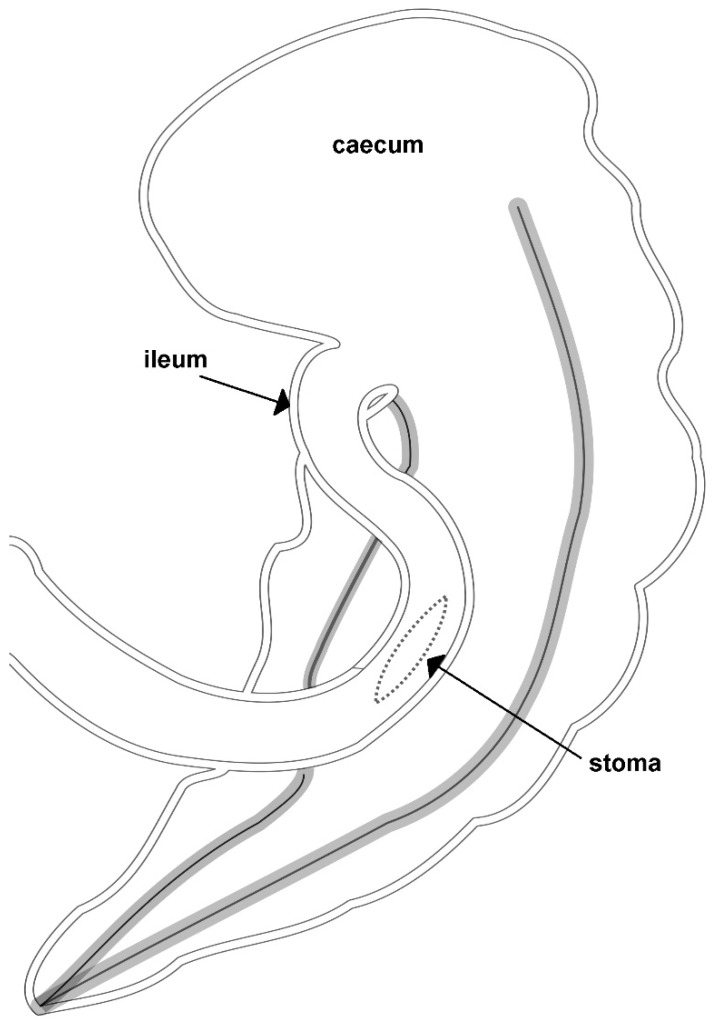
A representative illustration of an incomplete ileocecal bypass realized by Dr. Marco Gandini.

**Figure 2 animals-11-00403-f002:**
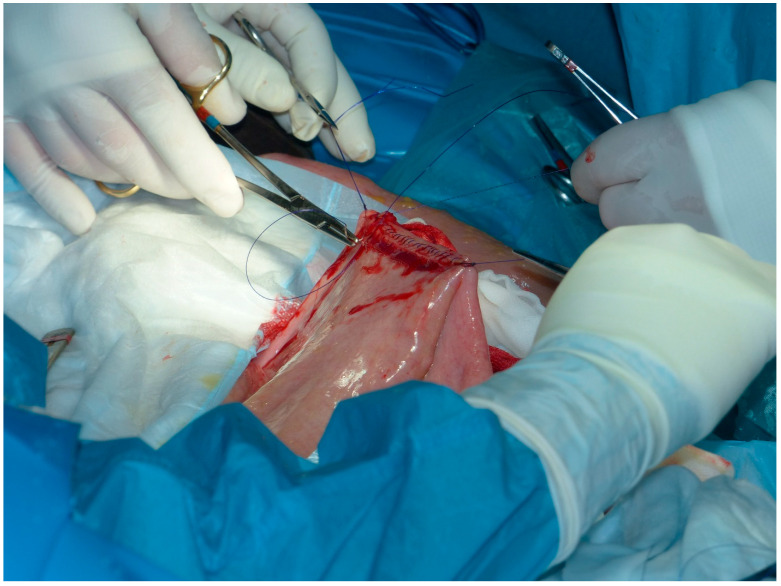
Continuous simple suture comprising mucosa and submucosa of jejunum and cecum applied to complete the underside of the anastomosis.

**Figure 3 animals-11-00403-f003:**
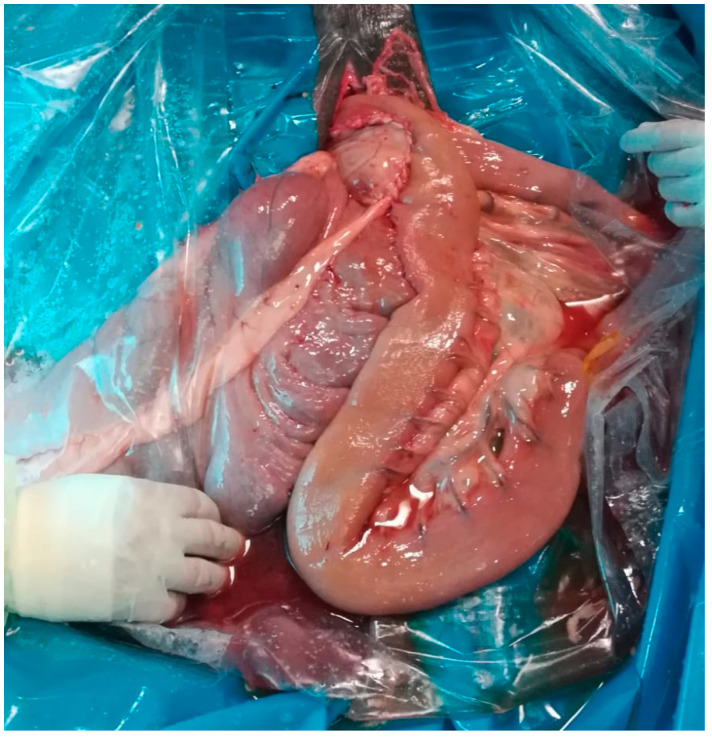
Incomplete ileocaecal bypass realized by hand-sewn technique.

**Figure 4 animals-11-00403-f004:**
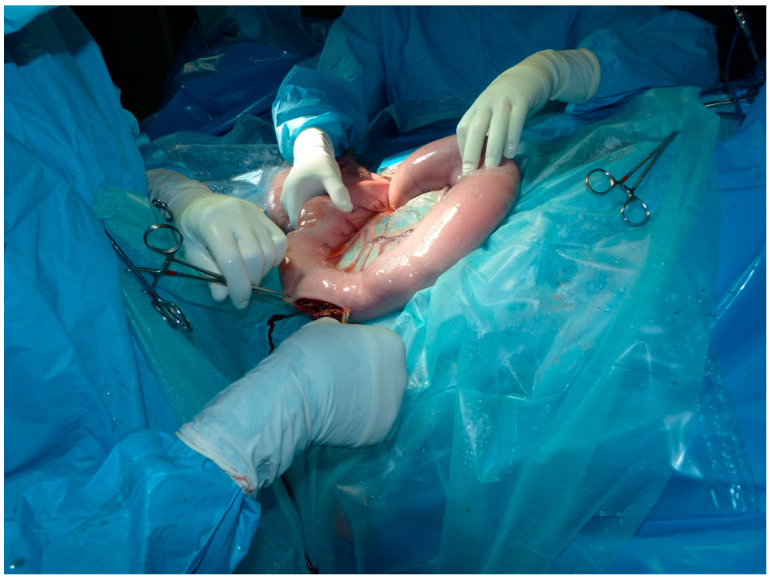
Enterotomy of the jejunum; the stoma was then used to connect the jejunum with the cecum.

## Data Availability

The data presented in this study are available on request from the corresponding author.

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
