# Peer review of "Incomplete Ileocecal Bypass for Ileal Pathology in Horses: 21 Cases (2012–2019)"

_animals, 2021, doi:10.3390/ani11020403_

Round 1

Reviewer 1 Report

The study represents a useful contribution to the scientific community of equine cases suffering from surgical disorders of the ileum, treated with an ileocecal bypass. This technique is not original, but large case series are missing in the literature, so a contribution with 21 cases is welcome.

Despite this, the paper must be improved to be accepted.  

Particularly, in the materials and methods section, some pivotal points are missing. Criteria adopted for intraoperative selection of patients submitted to the ileocecal bypass must be addressed, and some other parts of the procedure must be detailed as well (as enterotomy performed for ideal content evacuation).

The discussion must consider advantages and possible complications over other methods of treatment, comparing the results of previous studies, as control cases are missing. 

Many errors and punctuations omissions must be addressed in the references section.

A revision by an english mother tongue is strongly advised

Author Response

We would like to thank the Editor and reviewer for helping us improving our manuscript.

All the comments have been addressed and the whole manuscript has been submitted for proof reading to a professional service (please see attached certificate)

1)Particularly, in the materials and methods section, some pivotal points are missing. Criteria adopted for intraoperative selection of patients submitted to the ileocecal bypass must be addressed,

1) inserted in M&M

2)and some other parts of the procedure must be detailed as well (as enterotomy performed for ideal content evacuation).

2) inserted in M&M

3)The discussion must consider advantages and possible complications over other methods of treatment, comparing the results of previous studies, as control cases are missing. 

3) inserted, see text

Reviewer 2 Report

This is a very nice cases series, that I enjoyed reading.

I made a series of language corrections, but the discussion as a whole needs to be re-edited by a professional language editor who is familiar with veterinary terminology. Other than that, the MS is easy to read and nicely written.

The other important thing to edit is to clarify the first surgical technique described, I lost track after alignment of the intestinal track sections. 

The drawing and pictures are of excellent quality.

Please edit the following parts:

The language is easy to read, but written like spoken language. “the surgeon may be faced 9with the choice of whether to resect the intestinal tract

Simple summary

Please rephraise this sentence (“could “is not very scientific way to formulate a hypothesis)

”This latter technique could represent a valid alternative to extensive manipulation and it could reduce the recurrence of ileal occlusion and post-operative complication.”

Abstract

If short term survival is 93%, how can 12-month survival be 95%? Please clarify.

There is not really evidence to say “may reduce”, since there is nothing to compare to. Rephraise to “without subsequent” or something similar.

Introduction

Replace “colic” with “cases of colic” or something similar.

Line 40: in stead of “ the surgeon” use passive tense

Line 45: replace burden with infestation

Line 46: replace “our country” with the name of your country

Line 52: complication rate

M&M

Line 66: were these all complications that occurred or only ones, that were included in the study? Laminitis? Post-op fever etc?

Line 72: in which

Line 73: decision of

Line 75: lowercase benzylpenicillin, IU/kg, how much before the preoperative medication was given (range)?

Line 76: calcium gluconate is not universally used, please add indication

Line 77: Anaesthesia was induced with…

Line 78: …isoflurane in X% oxygen. (remove intubation remark)

Line 79: add dexamethasone route of administration, remove “just”.

Line 81: drape (not towel), remove “the”

Line 82: is->was

Line 88 confusing, what is distal part of the proximal third? Ileum? Picture is excellent, try to make the description super clear.

Figure 3: realized?

Line 112: please add suture material

Results

Line 116: stallions? Place in the order of frequency

Line 119: remove decimals

Line 124, remove decimals from PCV, TP7.2 +/-1.1, add unit

Note here, that previously commas were used and now dots, please unify the use of markings to journal style

Line 125: usually these are easier to read eg. “The mean amount of reflux was 0.4 litres (range 0–4 litres).”

Line 131: Do you mean, that in all cases the ileum was viable, but thickened and edematous?

Line 132: remove decimals

Line 132-133: poor language, please rewrite the centence and add mean.

Line 136 With

Line 140: If you have a culture result, please add it here

Line 146: functional might be a wrong choice of words here, since it is a postmortem finding

Line 148: Did any of the horses have hernias?

Line 158: Stubborn :)

Line 166: “Fecal test for Anoplocephala perfoliata was positive in 50% of the cases of ileal impaction reported in our study.” Please check journal guidelines, but usually latin names are written in italic font.

Line 167: dysfunction, relate? Please rephraise the whole sentence

Line 169: preferred->recommended, re-impaction->recurring impaction

Language check is needed.

Author Response

We would like to thank the Editor and reviewer for helping us improving our manuscript.

All the comments have been addressed and the whole manuscript has been submitted for proof reading to a professional service (please see attached certificate)

R1

1)Particularly, in the materials and methods section, some pivotal points are missing. Criteria adopted for intraoperative selection of patients submitted to the ileocecal bypass must be addressed,

1) inserted in M&M

2)and some other parts of the procedure must be detailed as well (as enterotomy performed for ideal content evacuation).

2) inserted in M&M

3)The discussion must consider advantages and possible complications over other methods of treatment, comparing the results of previous studies, as control cases are missing. 

3) inserted, see text

Rev 2

1)The other important thing to edit is to clarify the first surgical technique described, I lost track after alignment of the intestinal track sections. 

1) the description has been improved

 Simple summary

2)Please rephraise this sentence (“could “is not very scientific way to formulate a hypothesis)”This latter technique could represent a valid alternative to extensive manipulation and it could reduce the recurrence of ileal occlusion and post-operative complication.”

2) changed, see text 

Abstract

3)If short term survival is 93%, how can 12-month survival be 95%? Please clarify.

3) the reviewer is right. The meaning was that 95% of discharged horses survived.changed, see text

4)There is not really evidence to say “may reduce”, since there is nothing to compare to. Rephraise to “without subsequent” or something similar.

4) changed, see text

Introduction

5)Replace “colic” with “cases of colic” or something similar.

5) changed, see text

6)Line 40: in stead of “ the surgeon” use passive tense

6)Changed, see text

7)Line 45: replace burden with infestation

7) changed, see text

8)Line 46: replace “our country” with the name of your country

8) changed, see text

9) Line 52: complication rate

M&M

10) Line 66: were these all complications that occurred or only ones, that were included in the study? Laminitis? Post-op fever etc?

10) the reviewer is right. We added “fever”. We included only complications that could be related to the presence of the anastomosis

11) Line 72: in which

11) changed, see text

12) Line 73: decision of

12) changed, see text

13) Line 75: lowercase benzylpenicillin, IU/kg, how much before the preoperative medication was given (range)?

13) changed, see text

14) Line 76: calcium gluconate is not universally used, please add indication

14) inserted, see text

15)Line 77: Anaesthesia was induced with…

15) changed, see text

16)Line 78: …isoflurane in X% oxygen. (remove intubation remark)

16) changed, see text

17) Line 79: add dexamethasone route of administration, remove “just”.

17) changed, see text

18)Line 81: drape (not towel), remove “the”

18)changed, see text

19) Line 82: is->was

19)changed, see text

20) Line 88 confusing, what is distal part of the proximal third? Ileum? Picture is excellent, try to make the description super clear.

20) changed, see text

21)Figure 3: realized?

21) changed, see text

22)Line 112: please add suture material

22) changed, see text

Results

23)Line 116: stallions? Place in the order of frequency

23) changed, see text

24)Line 119: remove decimals

24) changed, see text

25)Line 124, remove decimals from PCV, TP7.2 +/-1.1, add unit

Note here, that previously commas were used and now dots, please unify the use of markings to journal style

25) changed, see text

26)Line 125: usually these are easier to read eg. “The mean amount of reflux was 0.4 litres (range 0–4 litres).”

26) changed, see text

27)Line 131: Do you mean, that in all cases the ileum was viable, but thickened and edematous?

27) changed, see text

28)Line 132: remove decimals

28) changed, see text

29)Line 132-133: poor language, please rewrite the centence and add mean.

29the manuscript has been proof-read by an editing service

30)Line 136 With

30) changed, see text

31)Line 140: If you have a culture result, please add it here

31) sorry, we didn’t took a swab on these cases

32)Line 146: functional might be a wrong choice of words here, since it is a postmortem finding

32)changed, see text

33)Line 148: Did any of the horses have hernias?

34) no

35)Line 158: Stubborn :)

35) changed, see text

36)Line 166: “Fecal test for Anoplocephala perfoliata was positive in 50% of the cases of ileal impaction reported in our study.” Please check journal guidelines, but usually latin names are written in italic font.

36) changed, see text

37)Line 167: dysfunction, relate? Please rephraise the whole sentence

37) changed, see text

38)Line 169: preferred->recommended, re-impaction->recurring impaction

38) changed, see text

Rev. 3

Abstract

1)Methods: additional detail on how the selected cases were analysed would be beneficial to include

1)Inserted, see text 

Introduction

2)Line 34: please amend to most colics and only a small percentage require

2) changed, see text

3)Line 36: not a new paragraph here, continue from previous sentence

3)changed, see text

4)Line 36: as a multispecies journal, suggesting adding ‘in the horse’ here

4) added, see text

5)Line 38: a brief outline of what the hypothetical explanations are would be beneficial

5) inserted, see text

6)Line 38: please remove while – not required

6)changed, see text

7)Line 52: please amend to complication

7) changed, see text

8) changed, see text

8)Line 56: I would present in past tense as your study is complete

Materials and Methods

9)I appreciate this is a case study review but your method should integrate a data analysis section outlining criteria for how you reviewed the cases and who conducted this to facilitate repetition of your approach.

9) inserted in M &M

10)Line 62-63: please provide abbreviations in full as 1st use in manuscript and not all readers may be familiar with these terms

10) changed, see text

11)Line 81: please amend to towels to avoid contamination – would consider replacing towels with surgical drapes?

11) changed, see text

12)Line 82: present in past tense, was rather than is – apply to description of surgery lines 87 to 113

12) changed, see text

Results

13)Again clearly presented, would be beneficial to present data as absolute values and also relative percentages to illustrate frequency in this sample and facilitate comparison to future studies

14) inserted, see text

14)Line 118: suggest amending to n=xx for number of cases in each breed

14) changed, see text

15)Line 123: suggest rephrasing to The average duration of abdominal presentation before admission was 12 hours (range 6-48 hours).

15) changed, see text

16)Line 124L add units to HR and TP

16) changed, see text

17)Line 125: final sentence is a little unclear, please edit to increase clarity

17) changed, see text

18) Line 125, 131: please double check your present figures as XX.X not XX,X

18) changed, see text

19)Line 133: suggest amending to were bypassed

19) changed, see text

20)Line 136: with spelt incorrectly

20) changed, see text

21)Line 141: seems little odd to present median here when using mean elsewhere

22) changed, see text

Discussion and conclusions

Line 177: insert the before ileum

Reviewer 3 Report

REVIEW Animals Incomplete ileocecal bypass for ileal pathology in horses

An interesting case study review related to equine colic which will be of interest and contribute to this field.

Simple summary

Nice summary provided; scope to simplify for non-veterinary readers who may not be familiar all terms included but recognise that given the subject matter reducing the veterinary terminology is challenging. Would be good to conclude with a summary sentence of what your findings suggest.

Line 9: would remove The as not needed in this sentence

Abstract

Clear summary and I like the use of headings to give the abstract increased structure

Methods: additional detail on how the selected cases were analysed would be beneficial to include

Introduction

Concise but relevant introduction to the topic which establishes a clear rationale for this study

Line 34: please amend to most colics and only a small percentage require

Line 36: not a new paragraph here, continue from previous sentence

Line 36: as a multispecies journal, suggesting adding ‘in the horse’ here

Line 38: a brief outline of what the hypothetical explanations are would be beneficial

Line 38: please remove while – not required

Line 52: please amend to complication

Line 56: I would present in past tense as your study is complete

Materials and Methods

Detailed explanation of selection of case and surgery presented. I appreciate this is a case study review but your method should integrate a data analysis section outlining criteria for how you reviewed the cases and who conducted this to facilitate repetition of your approach.

Line 62-63: please provide abbreviations in full as 1st use in manuscript and not all readers may be familiar with these terms

Line 81: please amend to towels to avoid contamination – would consider replacing towels with surgical drapes?

Line 82: present in past tense, was rather than is – apply to description of surgery lines 87 to 113

Results

Again clearly presented, would be beneficial to present data as absolute values and also relative percentages to illustrate frequency in this sample and facilitate comparison to future studies

Line 118: suggest amending to n=xx for number of cases in each breed

Line 123: suggest rephrasing to The average duration of abdominal presentation before admission was 12 hours (range 6-48 hours).

Line 124L add units to HR and TP

Line 125: final sentence is a little unclear, please edit to increase clarity

Line 125, 131: please double check your present figures as XX.X not XX,X

Line 133: suggest amending to were bypassed

Line 136: with spelt incorrectly

Line 141: seems little odd to present median here when using mean elsewhere

Discussion and conclusions

Relevant discussion and conclusions of the approaches taken and study limitations presented

Line 177: insert the before ileum

Author Response

We would like to thank the Editor and reviewer for helping us improving our manuscript.

All the comments have been addressed and the whole manuscript has been submitted for proof reading to a professional service (please see attached certificate)

R1

1)Particularly, in the materials and methods section, some pivotal points are missing. Criteria adopted for intraoperative selection of patients submitted to the ileocecal bypass must be addressed,

1) inserted in M&M

2)and some other parts of the procedure must be detailed as well (as enterotomy performed for ideal content evacuation).

2) inserted in M&M

3)The discussion must consider advantages and possible complications over other methods of treatment, comparing the results of previous studies, as control cases are missing. 

3) inserted, see text

Rev 2

1)The other important thing to edit is to clarify the first surgical technique described, I lost track after alignment of the intestinal track sections. 

1) the description has been improved

 Simple summary

2)Please rephraise this sentence (“could “is not very scientific way to formulate a hypothesis)”This latter technique could represent a valid alternative to extensive manipulation and it could reduce the recurrence of ileal occlusion and post-operative complication.”

2) changed, see text 

Abstract

3)If short term survival is 93%, how can 12-month survival be 95%? Please clarify.

3) the reviewer is right. The meaning was that 95% of discharged horses survived.changed, see text

4)There is not really evidence to say “may reduce”, since there is nothing to compare to. Rephraise to “without subsequent” or something similar.

4) changed, see text

Introduction

5)Replace “colic” with “cases of colic” or something similar.

5) changed, see text

6)Line 40: in stead of “ the surgeon” use passive tense

6)Changed, see text

7)Line 45: replace burden with infestation

7) changed, see text

8)Line 46: replace “our country” with the name of your country

8) changed, see text

9) Line 52: complication rate

M&M

10) Line 66: were these all complications that occurred or only ones, that were included in the study? Laminitis? Post-op fever etc?

10) the reviewer is right. We added “fever”. We included only complications that could be related to the presence of the anastomosis

11) Line 72: in which

11) changed, see text

12) Line 73: decision of

12) changed, see text

13) Line 75: lowercase benzylpenicillin, IU/kg, how much before the preoperative medication was given (range)?

13) changed, see text

14) Line 76: calcium gluconate is not universally used, please add indication

14) inserted, see text

15)Line 77: Anaesthesia was induced with…

15) changed, see text

16)Line 78: …isoflurane in X% oxygen. (remove intubation remark)

16) changed, see text

17) Line 79: add dexamethasone route of administration, remove “just”.

17) changed, see text

18)Line 81: drape (not towel), remove “the”

18)changed, see text

19) Line 82: is->was

19)changed, see text

20) Line 88 confusing, what is distal part of the proximal third? Ileum? Picture is excellent, try to make the description super clear.

20) changed, see text

21)Figure 3: realized?

21) changed, see text

22)Line 112: please add suture material

22) changed, see text

Results

23)Line 116: stallions? Place in the order of frequency

23) changed, see text

24)Line 119: remove decimals

24) changed, see text

25)Line 124, remove decimals from PCV, TP7.2 +/-1.1, add unit

Note here, that previously commas were used and now dots, please unify the use of markings to journal style

25) changed, see text

26)Line 125: usually these are easier to read eg. “The mean amount of reflux was 0.4 litres (range 0–4 litres).”

26) changed, see text

27)Line 131: Do you mean, that in all cases the ileum was viable, but thickened and edematous?

27) changed, see text

28)Line 132: remove decimals

28) changed, see text

29)Line 132-133: poor language, please rewrite the centence and add mean.

29the manuscript has been proof-read by an editing service

30)Line 136 With

30) changed, see text

31)Line 140: If you have a culture result, please add it here

31) sorry, we didn’t took a swab on these cases

32)Line 146: functional might be a wrong choice of words here, since it is a postmortem finding

32)changed, see text

33)Line 148: Did any of the horses have hernias?

34) no

35)Line 158: Stubborn :)

35) changed, see text

36)Line 166: “Fecal test for Anoplocephala perfoliata was positive in 50% of the cases of ileal impaction reported in our study.” Please check journal guidelines, but usually latin names are written in italic font.

36) changed, see text

37)Line 167: dysfunction, relate? Please rephraise the whole sentence

37) changed, see text

38)Line 169: preferred->recommended, re-impaction->recurring impaction

38) changed, see text

Rev. 3

Abstract

1)Methods: additional detail on how the selected cases were analysed would be beneficial to include

1)Inserted, see text 

Introduction

2)Line 34: please amend to most colics and only a small percentage require

2) changed, see text

3)Line 36: not a new paragraph here, continue from previous sentence

3)changed, see text

4)Line 36: as a multispecies journal, suggesting adding ‘in the horse’ here

4) added, see text

5)Line 38: a brief outline of what the hypothetical explanations are would be beneficial

5) inserted, see text

6)Line 38: please remove while – not required

6)changed, see text

7)Line 52: please amend to complication

7) changed, see text

8) changed, see text

8)Line 56: I would present in past tense as your study is complete

Materials and Methods

9)I appreciate this is a case study review but your method should integrate a data analysis section outlining criteria for how you reviewed the cases and who conducted this to facilitate repetition of your approach.

9) inserted in M &M

10)Line 62-63: please provide abbreviations in full as 1st use in manuscript and not all readers may be familiar with these terms

10) changed, see text

11)Line 81: please amend to towels to avoid contamination – would consider replacing towels with surgical drapes?

11) changed, see text

12)Line 82: present in past tense, was rather than is – apply to description of surgery lines 87 to 113

12) changed, see text

Results

13)Again clearly presented, would be beneficial to present data as absolute values and also relative percentages to illustrate frequency in this sample and facilitate comparison to future studies

13) inserted, see text

14)Line 118: suggest amending to n=xx for number of cases in each breed

14) changed, see text

15)Line 123: suggest rephrasing to The average duration of abdominal presentation before admission was 12 hours (range 6-48 hours).

15) changed, see text

16)Line 124L add units to HR and TP

16) changed, see text

17)Line 125: final sentence is a little unclear, please edit to increase clarity

17) changed, see text

18) Line 125, 131: please double check your present figures as XX.X not XX,X

18) changed, see text

19)Line 133: suggest amending to were bypassed

19) changed, see text

20)Line 136: with spelt incorrectly

20) changed, see text

21)Line 141: seems little odd to present median here when using mean elsewhere

22) changed, see text

Discussion and conclusions

Line 177: insert the before ileum

Round 2

Reviewer 1 Report

The paper is much improved, and can be published in the present form